# Metabolomic signatures associated with depression and predictors of antidepressant response in humans: A CAN-BIND-1 report

Giorgia Caspani [1], Gustavo Turecki [2,3], Raymond W. Lam [4], Roumen V. Milev [5,6], Benicio N. Frey [7,8], Glenda M. MacQueen[9], Daniel J. Müller[10,11], Susan Rotzinger[11,12,13], Sidney H. Kennedy[11,12,13], Jane A. Foster [7,8,13] & Jonathan R. Swann [1,14✉]

One of the biggest challenges in treating depression is the heterogeneous and qualitative nature of its clinical presentations. This highlights the need to find quantitative molecular markers to tailor existing treatment strategies to the individual's biological system. In this study, high-resolution metabolic phenotyping of urine and plasma samples from the CAN-BIND study collected before treatment with two common pharmacological strategies, escitalopram and aripiprazole, was performed. Here we show that a panel of LDL and HDL subfractions were negatively correlated with depression in males. For treatment response, lower baseline concentrations of apolipoprotein A1 and HDL were predictive of escitalopram response in males, while higher baseline concentrations of apolipoprotein A2, HDL and VLDL subfractions were predictive of aripiprazole response in females. These findings support the potential of metabolomics in precision medicine and the possibility of identifying personalized interventions for depression.

[1] Department of Metabolism, Digestion and Reproduction, Imperial College London, London, UK. [2] Department of Psychiatry, McGill University, Montreal, QC, Canada. [3] Douglas Mental Health University Institute Frank B. Common (FBC), Montreal, QC, Canada. [4] Department of Psychiatry, University of British Columbia, Vancouver, BC, Canada. [5] Department of Psychiatry, Queen's University, Kingston, ON, Canada. [6] Providence Care Hospital, Kingston, ON, Canada. [7] Department of Psychiatry & Behavioral Neurosciences, McMaster University, Hamilton, ON, Canada. [8] St. Joseph's Healthcare, Hamilton, ON, Canada. [9] Hotchkiss Brain Institute, University of Calgary, Calgary, AL, Canada. [10] Pharmacogenetics Research Clinic, Centre for Addiction and Mental Health, Toronto, ON, Canada. [11] Department of Psychiatry, University of Toronto, Toronto, ON, Canada. [12] Department of Psychiatry, Centre for Mental Health, University Health Network, Krembil Research Institute, University of Toronto, Toronto, ON, Canada. [13] Centre for Depression and Suicide Studies, St Michael's Hospital, Li Ka Shing Knowledge Institute, Toronto, ON, Canada. [14] School of Human Development and Health, Faculty of Medicine, University of Southampton, Southampton, UK. ✉email: j.swann@soton.ac.uk

Major depressive disorder (MDD) is a complex, debilitating psychiatric disorder. Not only is it the most prevalent mood disorder, but it is also the leading cause of disability worldwide, with an estimated 322 million people affected globally[1]. MDD is a heterogeneous disease characterized by highly variable symptomatic manifestations[2]. Our ability to elicit the biology of MDD is impacted by the lack of association between clinical symptoms and underlying biology. Currently, treatment selection is based on a trial-and-error strategy[3]. As a result, the proportion of patients who achieve complete remission after a first pharmacological therapy is limited to one third[4], while ~30% of patients have not achieved remission after four consecutive treatments[5,6]. These observations highlight the urgent need to identify biomarkers of MDD and predictors of treatment response, to refine diagnosis, and personalization of treatment strategies to enhance their effectiveness.

Metabolic phenotyping involves the comprehensive measurement of metabolites in biological samples to identify disease-associated metabolic signatures and to characterize biochemical responses to stimuli. This powerful technology enables a broad range of biochemical events occurring in the whole organism to be studied simultaneously[7]. Previous metabolomic studies have identified metabolic signatures of MDD in both urine and plasma, including disruptions to amino acid and energy metabolism as well as alterations in ketone bodies and gut microbial metabolites[8,9]. Pharmacometabolomics has emerged as a promising approach that can predict an individual's response to pharmacological therapy based on their pre-dose metabolic phenotype. For example, individuals who responded to sertraline or placebo treatment could be discriminated from those who did not by investigating their serum metabolic profile at baseline[10] and its change over time[11]. Baseline plasma lipids such as phosphatidylcholines and non-hydroxylated sphingomyelins[12], as well as metabolites from the tryptophan, tyrosine, and purine pathways[13], were predictive of response to citalopram/escitalopram and to the combination of bupropion and escitalopram. Similarly, changes in glutamate and circulating phospholipids correlated with responses to ketamine and its enantiomer esketamine[14], while glycine was negatively associated with escitalopram treatment outcome[15]. Both sex and BMI are likely to exert a strong influence on pharmacokinetics. There have been reports of females being more responsive to selective serotonin reuptake inhibitors (SSRIs) than males[16], possibly due to hormones like progesterone and estradiol, lower rates of gastric emptying (leading to quicker absorption[17]), and higher adiposity (leading to greater distribution[18,19]). For this reason, the present study investigated sex differences in metabolic associations with treatment outcome, while accounting for the confounding effects of BMI, as well as age and inflammation. Although there is extensive evidence supporting a link between dyslipidemia and MDD[20–24], to the best of our knowledge, only one metabolomic study to date has defined differences in lipoprotein profile between MDD and healthy control (HC) participants[25], and none has tested the potential of lipoproteins to predict treatment outcome.

Canadian Biomarker Integration Network in Depression (CAN-BIND) is a national network of researchers across eight Canadian universities that focuses on identifying biomarkers of depression and their potential to predict treatment response[26]. In this study, high-resolution metabolic phenotyping of urine and plasma samples from the CAN-BIND study collected pre-treatment with two common pharmacological strategies, escitalopram and aripiprazole, was performed. Significant, sex-specific metabolic signatures of depression and of antidepressant treatment response were identified in plasma, but not in urine. In males, depression was significantly correlated with low plasma concentrations of a number of LDL and HDL subfractions. Additionally, low baseline concentrations of apolipoprotein A1 and HDL were predictive of escitalopram response in males, while high baseline concentrations of apolipoprotein A2, HDL, and VLDL subfractions were predictive of aripiprazole augmentation response in females.

## Results

**Clinical characteristics of the CAN-BIND 1 cohort.** A total of 323 participants were recruited from 6 outpatient centers across Canada between August 2013 and December 2016[26–28]. These consisted of 112 HCs (Montgomery–Åsberg Depression Rating Scale [MADRS] $= 0.8 \pm 1.7$) and 211 depressed (MDD) participants (MADRS $= 29.9 \pm 5.6$). The groups were matched for age ($N = 323$, Mann–Whitney $U = 17,229.5$, $p = 0.252$) and sex ($N = 323$, $\chi(1) = 0.0041$; $p = 0.9491$). Participants with clinical depression underwent a 16-week two-phase treatment trial: escitalopram (10–20 mg) was administered during phase I (baseline to week 8), and escitalopram either alone (in responders) or augmented with aripiprazole (2–10 mg, in non-responders) was administered during phase II (week 9–16), as shown in Fig. 1. Plasma and urine samples were collected pre-treatment and analyzed by $^1$H nuclear magnetic resonance (NMR), giving rise to a panel of 112 lipoproteins (Supplementary Data 1) and 50 urinary metabolites (Supplementary Data 2). PCA models constructed on the urine metabolic profiles and plasma lipoprotein profiles of all participants showed that no clustering was apparent based on the site that the samples were collected from, demonstrating that the metabolic profile was not affected by study location (Supplementary Fig. 1). Plasma data from 12 participants and urine data from 9 participants were excluded as they did not pass QC checks. In addition, 31 participants dropped out of the study before the termination of the clinical trial. The $N$ given in the following sections reflects the number of samples that were included in the analysis, excluding removed observations and dropouts. Of the remaining 180 participants, 83 (46%) were given escitalopram monotherapy for the entire duration of the study, while 97 (54%) were augmented with aripiprazole in phase II. Further demographic and clinical information on the participants, including response rates, can be found in Table 1.

**Biochemical signature of MDD.** To determine the biochemical variation associated with depression, weighted correlation networks were constructed on the metabolic phenotypes of MDD and HC participants. Correlation network analysis allows the relationship of clusters of highly associated features with an outcome of interest to be studied. In this section, separate networks were built for the plasma and urinary data and their metabolic associations with MADRS score at baseline, as well as with the covariates age, BMI, and CRP. The analytical workflow is depicted in Fig. 1.

In the plasma data ($N = 310$), 10 distinct modules were identified with high topological overlap ($\beta = 7$). Of these, only module 1, containing six highly correlated metabolites, was positively associated with MADRS score at baseline ($p = 0.004$), as well as with age ($p < 0.001$) and BMI ($p < 0.001$), (Supplementary Fig. 2a). This included low-density lipoprotein (LDL)$_6$ and its subfractions (triglycerides, cholesterol, free cholesterol, phospholipids, apolipoprotein B).

In total, seven modules were positively associated with age, indicating a general age-related increase in lipoprotein concentrations. For CRP, four modules (LDL of high size and very-low-density lipoproteins [VLDL]-predominant) were positively associated and four (high-density lipoproteins [HDL]-predominant, and LDL of low size) were negatively associated, indicating the

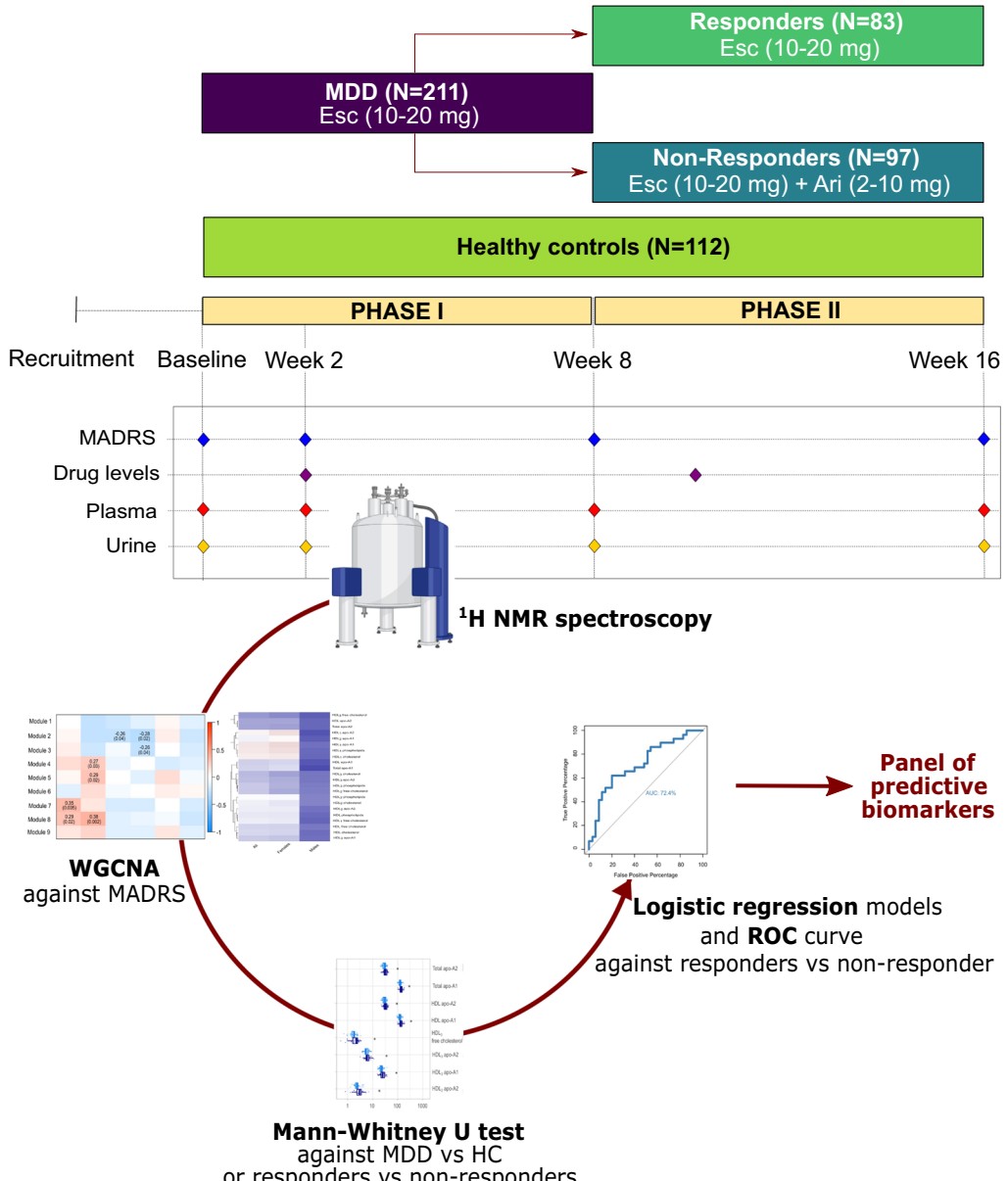

**Fig. 1 Design of the CAN-BIND 1 study and analytical workflow.** In phase I (week 0–8), MDD participants ($N = 211$) received escitalopram (ESC). MDD participants who responded to therapy continued on this treatment ($N = 83$), while non-responders received an augmentation with aripiprazole (ARI) during phase II (week 9–16, $N = 97$). Red diamonds show the timings of data collection for assessment of depressive symptoms (via Montgomery–Åsberg Depression Rating Scale [MADRS] at baseline, week 2, week 8, week 16), circulating drug levels for both ESC and ARI (week 2 and week 10) and blood and urine samples (only baseline data considered). Samples were analyzed by proton nuclear magnetic resonance ($^1$H NMR) spectroscopy (spectrometer image created by www.bioRender.com) and the resulting plasma and urine data were studied by weighted gene correlation network analysis (WGCNA), and significant features were validated by univariate statistics (Mann–Whitney $U$ test) and by the generation of receiver operating characteristic (ROC) curves. Significant features met the following criteria: (1) exhibited WGCNA $p < 0.05$, (2) have a $p < 0.05$ in Mann–Whitney U test, (3) pass the Benjamini–Hochberg correction at an false detection rate of 20%.

contrasting relationships between high-density lipoproteins (HDLs) and LDLs and the immune system and inflammation. This information is summarized in Supplementary Fig. 2a.

Considering males and females independently, additional significant associations were highlighted (Supplementary Fig. 2b, c). In males ($N = 115$, 1 outlier removed, $\beta = 2$), module 4 ($p = 0.015$) and module 6 ($p < 0.001$) were negatively correlated with MADRS at baseline. The former was also negatively associated with BMI ($p = 0.004$) and CRP ($p = 0.036$). In females ($\beta = 5$), module 7 was positively associated with baseline MADRS score ($p = 0.016$) and with age ($p = 0.004$), BMI ($p < 0.001$), and CRP ($p = 0.014$). In both

sexes, additional significant correlations were found between modules and the covariates, as shown in Supplementary Fig. 2b, c. The correlations between each metabolite and the baseline MADRS score for all participants, as well as for the male and female participants, is depicted in the heatmap in Fig. 2a. Based on the Mann–Whitney $U$ test after the FDR correction, baseline concentrations of HDL-cholesterol ($p = 0.017$), HDL$_2$-cholesterol ($p = 0.027$), LDL$_2$ ($p = 0.003$), LDL$_2$-cholesterol ($p = 0.001$), LDL$_2$-free cholesterol ($p = 0.003$), LDL$_3$-free cholesterol ($p = 0.037$), LDL$_2$-phospholipids ($p = 0.001$), LDL$_2$-apolipoprotein B ($p = 0.003$) were found to be significantly lower in males with depression compared

**Table 1 Patients' demographic and clinical information.**

| | Full cohort (N = 323) | Control (N = 112) | MDD (N = 211) |
|---|---|---|---|
| *Sex* | | | |
| Males | 37% | 37% | 37% |
| Females | 63% | 63% | 63% |
| Age | 34.5 ± 12.1 years | 33.0 ± 10.7 years | 35.3 ± 12.6 years |
| BMI | 25.8 ± 6.0[a] | 24.3 ± 4.7[b] | 26.5 ± 6.4[b] |
| Non-obese (<29.9) | 78% | 90% | 72% |
| Obese (>30) | 22% | 10% | 28% |
| MADRS | 19.8 ± 14.6 | 0.8 ± 1.7 | 29.9 ± 5.6 |
| *Admission site* | | | |
| Centre for Addiction and Mental Health (University of Toronto) | 5% | 6% | 4% |
| McMaster University | 16% | 17% | 15% |
| Queen's University | 11% | 14% | 10% |
| University Health Network (Toronto General/Western Hospital) | 24% | 21% | 25% |
| University of British Columbia | 23% | 11% | 30% |
| University of Calgary | 21% | 31% | 16% |
| *Drug regime* | | | |
| ESC + ESC | | | 46%[c] |
| ESC + ARI augmentation | | | 54%[c] |
| Responders at week 8 | | | 47%[c] |
| Responders at week 16 | | | 75%[d] |
| to ESC | | | 92%[a] |
| to ARI | | | 59%[e] |
| Dropouts | | | 15% |

Information about the participants is summarized below for the full cohort, as well as for healthy controls and depressed (MDD) individuals.
[a]6 missing datapoints.
[b]3 missing datapoints.
[c]31 missing datapoints.
[d]45 missing datapoints.
[e]9 missing datapoints.

week 16), as well as the covariates of interest (i.e. age, BMI and CRP).

To identify metabolic predictors of response to escitalopram monotherapy, the correlation between baseline plasma metabolic signatures and the percentage reduction in MADRS score during phase I was evaluated in all participants ($\beta = 5$). Eleven modules of highly correlated plasma metabolites were identified (Supplementary Fig. 3a). However, none of these modules were significantly correlated with percentage MADRS reduction after phase I. Stratifying by sex (Supplementary Fig. 3b, c) highlighted two modules (module 2 [$p = 0.022$]; module 3 [$p = 0.036$]) negatively correlated with MADRS reduction in males ($N = 64$, one outlier removed, $\beta = 6$). Module 2 was also negatively associated with CRP ($p = 0.037$), while module 3 was not correlated with any covariates. No significant associations were detected in females ($N = 104$, three outliers removed, $\beta = 6$). The significant correlations with MADRS reduction in phase I are depicted in the heatmap in Fig. 3a. A Mann–Whitney U test was performed on metabolites belonging to the significant modules to confirm differences between responders and non-responders to escitalopram. Apolipoprotein A1 ($p = 0.012$) and A2 ($p = 0.035$), HDL-apolipoprotein A1 ($p = 0.024$) and A2 ($p = 0.047$), HDL$_3$-free cholesterol ($p = 0.031$), HDL$_3$-apolipoprotein A1 ($p = 0.025$), HDL$_2$-apolipoprotein A2 ($p = 0.023$), and HDL$_3$-apolipoprotein A2 ($p = 0.038$) were significantly lower in male MDD participants at baseline who responded to escitalopram compared to the non-responders (Fig. 3b). The pre-treatment and post-treatment fold change of the metabolites belonging to the significant modules is shown in Supplementary Data 3 for both male responders and non-responders. Spearman correlation was performed between the baseline concentration of these metabolites and the circulating norescitalopram/escitalopram ratio at week 2, but no significant associations were identified (Fig. 3c). The ROC curve built on this panel of 8 plasma metabolites in males had an AUC of 72.4%, CI: 0.5974–0.8509 (Fig. 3d). From the logistic regression analysis, apolipoprotein A1 (Rho = $-0.282$, $p = 0.036$), HDL-apolipoprotein A1 (Rho = $-0.267$, $p = 0.046$), and HDL$_3$-free cholesterol (Rho = $-0.293$, $p = 0.028$) showed a significant negative correlation with a reduction in MADRS in phase I after correcting for age, BMI and CRP.

to the control males (Fig. 2b). No significant differences were identified in females ($N = 190$, four outliers removed). A partial correlation between metabolic concentrations and MADRS score corrected for age, BMI and CRP further confirmed that LDL$_2$ (Rho = $-0.192$, $p = 0.046$), LDL$_2$-cholesterol (Rho = $-0.190$, $p = 0.049$), LDL$_2$-phospholipids (Rho = $-0.202$, $p = 0.036$) and LDL$_2$-apolipoprotein B (Rho = $-0.192$, $p = 0.046$) were significantly inversely correlated with depression severity independently of the covariates.

The weighted correlation network constructed on the urinary metabolites from all participants ($N = 314$) indicated no association between any of the five modules and MADRS at baseline. Similarly, no significant correlations were found after stratifying by sex. Due to a lack of significant correlations, no further analysis on urinary data was carried out.

**Metabolic predictors of escitalopram response**. To investigate the ability of baseline plasma metabolites to predict later improvement in depressive symptoms, weighted correlation networks were constructed exclusively on MDD participants ($N = 166$, six outliers removed). In the following sections, networks were constructed to investigate the association between metabolic clusters and the percentage reduction in MADRS score during phase I (between baseline and week 8), phase II (between week 8 and week 16), and during the entire trial (between baseline and

**Metabolic predictors of aripiprazole augmentation response**. Weighted correlation networks were built on the plasma profiles of MDD participants receiving aripiprazole augmentation ($N = 73$, five outliers removed) to identify baseline metabolites associated with the percentage reduction in MADRS score during phase II. From the plasma profiles of all MDD participants ($\beta = 5$), 11 modules were identified and 7 were positively associated with a reduction in MADRS in phase II (Supplementary Fig. 3a). Stratifying by sex indicated that this observation was driven by changes in the females ($N = 43$, two outliers removed, $\beta = 6$), with no baseline lipoproteins associated with MADRS change in the males ($N = 32$, one outlier removed, $\beta = 6$) (Supplementary Fig. 3b). Modules 2 ($p = 0.008$), 3 ($p < 0.001$), 5 ($p = 0.002$), 7 ($p = 0.004$), 9 ($p < 0.001$), and 10 ($p < 0.001$) were positively correlated with a reduction in MADRS during phase II. The majority of these were also positively associated with age and BMI, as evident in Supplementary Fig. 3c. The correlations between individual metabolites and phase II MADRS reduction are shown in the heatmap in Fig. 4a. Of the 71 metabolites belonging to these modules, baseline concentrations of 26 lipoproteins were found to be significantly higher in female participants who responded to aripiprazole augmentation at week 16 than in those who did not (Fig. 4b). These lipoproteins were apolipoprotein A2 ($p = 0.004$), LDL$_5$ ($p = 0.030$), intermediate-density lipoprotein

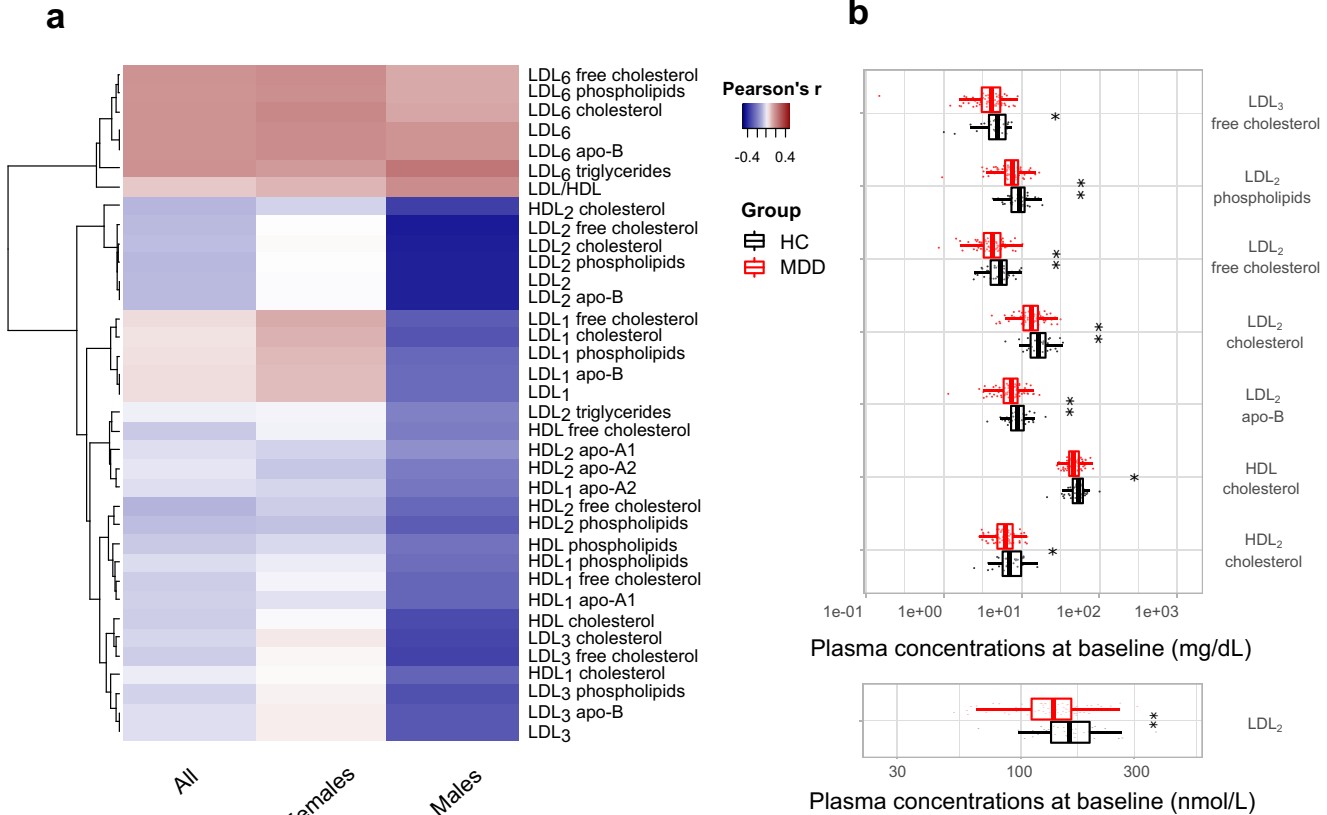

**Fig. 2 Biochemical signature of major depressive disorder (MDD). a** Heatmap depicting the correlation coefficients between the baseline concentrations of plasma metabolites in the significant weighted correlation network analysis (WGCNA) modules and baseline MADRS score for all ($N = 310$), female ($N = 190$) and male participants ($N = 115$). Positive correlations are shown in red and negative correlations are shown in blue. **b** Lipoproteins from the WGCNA analysis that are statistically significantly different between MDD and healthy control (HC) males. Boxplots show median (with first and third quartile), bars indicate 95% confidence interval of the median. Statistical significance determined by Mann–Whitney $U$ test after false discovery rate correction (*$p < 0.05$, **$p < 0.01$, ***$p < 0.001$).

(IDL)-triglycerides ($p = 0.050$), HDL-apolipoprotein A2 ($p = 0.003$), VLDL$_2$-triglycerides ($p = 0.034$), VLDL$_3$-triglycerides ($p = 0.032$), VLDL$_2$-cholesterol ($p = 0.021$), VLDL$_3$-cholesterol ($p = 0.023$), VLDL$_4$-cholesterol ($p = 0.019$), VLDL$_4$-free cholesterol ($p = 0.020$), VLDL$_2$-phospholipids ($p = 0.038$), VLDL$_3$-phospholipids ($p = 0.040$), LDL$_5$-cholesterol ($p = 0.046$), LDL$_5$-free cholesterol ($p = 0.032$), LDL$_5$-phospholipids ($p = 0.042$), LDL$_5$-apolipoprotein B ($p = 0.030$), HDL$_3$-cholesterol ($p = 0.034$), HDL$_4$-cholesterol ($p = 0.043$), HDL$_3$-free cholesterol ($p = 0.015$), HDL$_4$-free cholesterol ($p = 0.002$), HDL$_3$-phospholipids ($p = 0.026$), HDL$_4$-phospholipids ($p = 0.006$), HDL$_3$-apolipoprotein A1 ($p = 0.036$), HDL$_4$-apolipoprotein A1 ($p = 0.038$), HDL$_3$-apolipoprotein A2 ($p = 0.010$), and HDL$_4$-apolipoprotein A2 ($p = 0.012$). Individual fold changes in these lipoprotein features from week 16 to baseline are reported in Supplementary Data 4. Baseline concentrations of LDL$_5$ (Rho = 0.406, $p = 0.009$), as well as its phospholipid (Rho = 0.373, $p = 0.017$), cholesterol (Rho = 0.374, $p = 0.016$), and apolipoprotein B (Rho = 0.406, $p = 0.009$) subfractions, were found to be positively correlated with dehydroaripiprazole/aripiprazole ratio at week 16 (Fig. 4c), suggesting more extensive aripiprazole metabolism in the responders. The top 10 lipoproteins (by $p$ value) associated with response to the combination of escitalopram and aripiprazole were used to construct a logistic regression model. The resulting ROC curve showed that this panel of baseline metabolites (HDL$_4$-free cholesterol, HDL-apolipoprotein A2, apolipoprotein A2, HDL$_4$-phospholipids, HDL$_3$-apolipoprotein A2, HDL$_4$-apolipoprotein

A2, HDL$_3$-free cholesterol, VLDL$_4$-cholesterol, VLDL$_4$-free cholesterol, VLDL$_2$-cholesterol) at baseline could discriminate female responders from non-responders to adjunctive aripiprazole therapy at week 16 with an accuracy of 88.5%, CI: 0.7922–0.9776 (Fig. 4d). Furthermore, apolipoprotein A2 (Rho = 0.360, $p = 0.040$), HDL-apolipoprotein A2 (Rho = 0.383, $p = 0.040$), HDL$_3$-free cholesterol (Rho = 0.361, $p = 0.040$) and HDL$_3$-apolipoprotein A2 (Rho = 0.448, $p = 0.026$) were positively correlated with the percentage reduction in MADRS score occurring in phase II after controlling for age, BMI and CRP.

## Discussion

With the increasing burden of MDD, its heterogenous symptomology, and inter-individual variation in treatment responses, there is a pressing need to identify clinically relevant non-invasive biomarkers for diagnosing MDD and informing personalized pharmacological strategies. In this work, we have identified sex-specific biochemical signatures of depression in the plasma lipoprotein profiles as well as predictive markers of escitalopram and aripiprazole response. For MDD, several lipoprotein species were significantly lower in the circulation of male MDD participants compared to their control equivalents. This included the overall number of LDL$_2$ particles and their main fractions, cholesterol and free cholesterol, apolipoprotein B, and phospholipids. In addition, HDL-cholesterol and HDL$_2$-cholesterol were less abundant in the circulation of MDD males. This is consistent with previous reports of a link between low cholesterol and

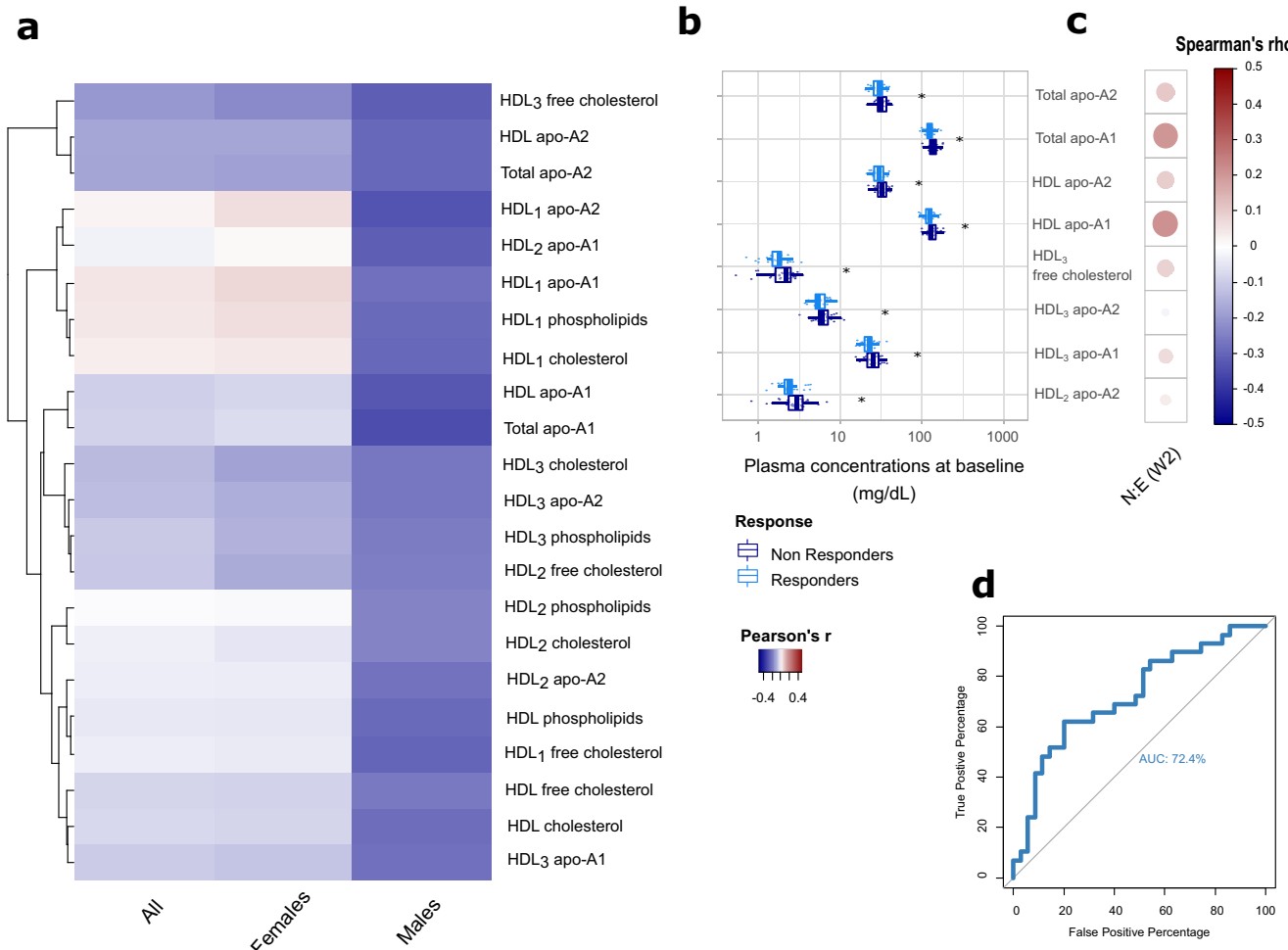

**Fig. 3 Biochemical predictors of response to escitalopram monotherapy. a** Heatmap depicting the correlation coefficients between the baseline concentrations of plasma metabolites in the WGCNA modules significantly associated with the reduction in MADRS score during phase I in all ($N = 166$), female ($N = 104$) and male participants ($N = 64$). In red, metabolites that display a positive association with MADRS reduction, in blue those that display a negative association (i.e. low baseline concentrations predictive of improvement (greater reduction)). **b** Baseline lipoproteins that are significantly different between responders and non-responders to escitalopram at week 8 in males (Mann–Whitney $U$ test with false discovery rate; *$p < 0.05$, **$p < 0.01$, ***$p < 0.001$). Boxplots show median (with first and third quartile), bars indicate 95% confidence interval of the median. **c** Correlogram showing the correlation coefficients (positive in red; negative in blue; Spearman correlation) between baseline lipoproteins and circulating norescitalopram/escitalopram ratio at week 2. No significant associations were detected. **d** Receiver operating characteristic (ROC) curve evaluating the ability of apolipoprotein A1, apolipoprotein A2, HDL apolipoprotein A1, HDL apolipoprotein A2, HDL$_3$ free cholesterol, HDL$_3$ apolipoprotein A1, HDL$_2$ apolipoprotein A2, HDL$_3$ apolipoprotein A2 at baseline to discriminate between male escitalopram responders and non-responders at week 8 (AUC = 72.4%).

depression[20–22]. Cholesterol depletion in the CNS affects central nerve terminal structure and function and modulates receptor responsiveness to serotonin. However, it remains unclear if peripheral cholesterol can impact on central serotonergic neurotransmission[29]. The present study revealed that LDL can have opposing relationships depending on size and density. LDL$_6$, which is small (21.8–23.2 nm) and dense, was positively associated with MADRS, while LDL$_2$, which is larger (25.5–26.0 nm) and less dense, was negatively associated with MADRS. Size and density confer LDL lipoproteins with specific electrical, immunoreactive, and biochemical properties[30]. Small dense LDL (known as 'pattern B'), as opposed to larger less dense LDL particles ('pattern A'), is related to cardiovascular risk[31] due to their lower affinity for the LDL receptor, resulting in slower LDL clearance from the circulation[32]. Interestingly, there are strong links between MDD and coronary heart disease and the prognosis for both worsens when present together[33,34]. This relationship was not observed in the female participants. Inherent sex differences in lipoprotein metabolism and the wide age range including

both pre- and post-menopausal females (a process known to alter lipid homeostasis[35]) may underlie these observations.

Clear sex differences were also observed in relation to lipoprotein signatures of antidepressant response. In males, but not females, several HDL subfractions were predictive of treatment response to escitalopram. Conversely, a broad range of lipoproteins (VLDL, LDL, IDL, HDL) was predictive of response to aripiprazole augmentation in females but not males. For escitalopram, male responders had low baseline concentrations of HDL subfractions. These subfractions were also noted to be negatively correlated with MADRS in the males suggesting that individuals with severe depressive symptoms had a greater response to escitalopram. This is consistent with the observation that antidepressant drugs are less effective for patients with mild depression compared to those with moderate to severe symptoms[36]. Low baseline concentrations of apolipoprotein A1, HDL-apolipoprotein A1, and HDL$_3$-free cholesterol were also predictive of escitalopram response 8 weeks later even after controlling for age, BMI, and inflammation (determined by CRP).

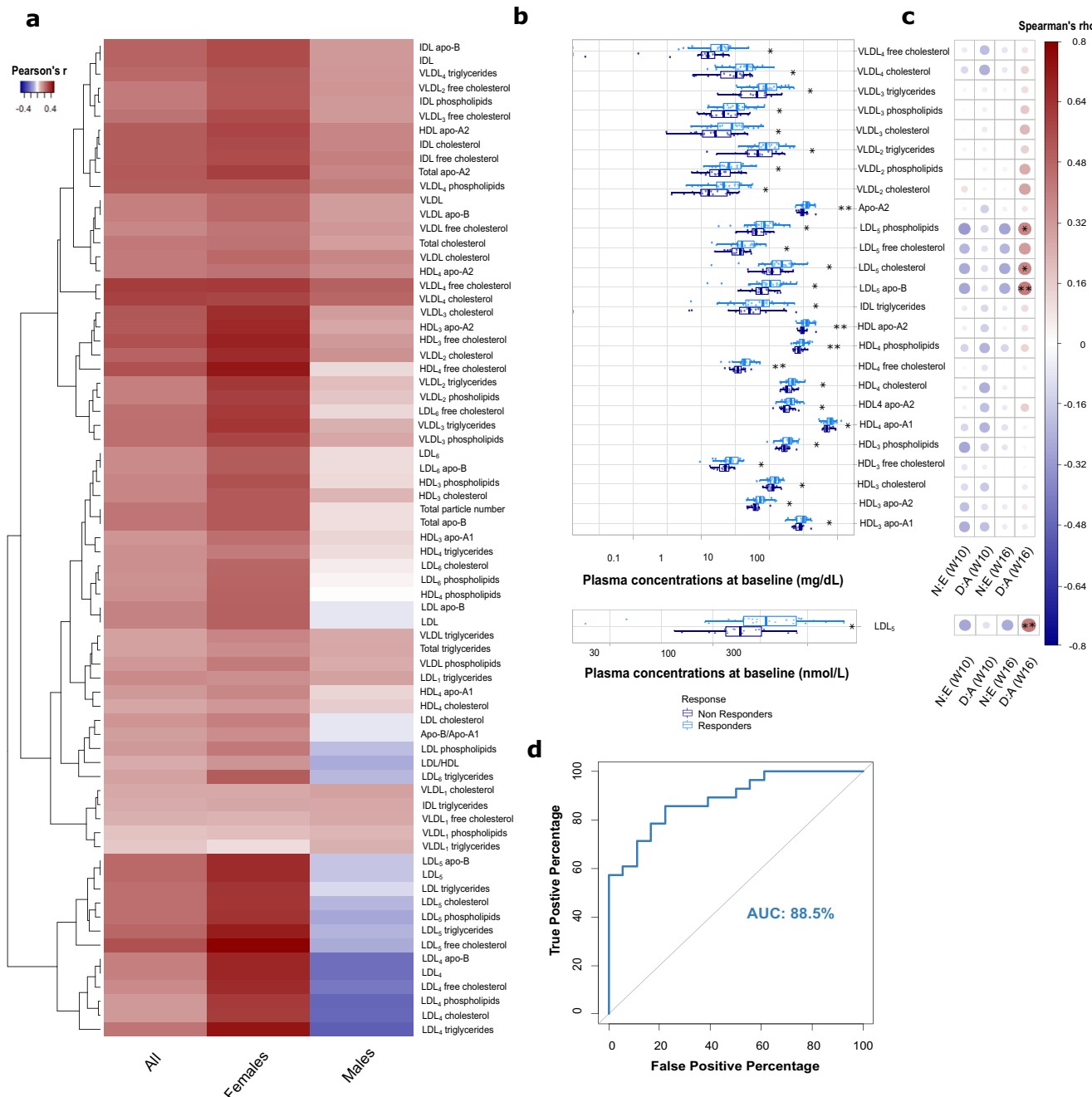

**Fig. 4 Biochemical predictors of response to escitalopram and aripiprazole combined therapy. a** Heatmap identifying the correlation coefficients between the baseline concentrations of plasma metabolites in the significant WGCNA modules and the reduction in MADRS score during phase II in all (N = 73), female (N = 43) and male participants (N = 32). In red, metabolites that display a positive association with MADRS reduction, in blue those that display a negative association (e.g. low baseline concentrations predictive of improvement). **b** Baseline lipoproteins that are significantly different between responders and non-responders to aripiprazole at week 16 in females (Mann–Whitney U test with false discovery rate; *p < 0.05, **p < 0.01, ***p < 0.001). Boxplots show median (with first and third quartile), bars indicate 95% confidence interval of the median. **c** Correlogram depicting the correlation coefficients (positive in red; negative in blue; Spearman correlation) between baseline lipoproteins and circulating norescitalopram/escitalopram (N:E) and dehydroaripiprazole/aripirazole (D:A) ratios at weeks 2, 10, and 16. **d** Receiver operating characteristic (ROC) curve evaluating the ability of baseline HDL$_4$ free cholesterol, HDL apolipoprotein-A2, total apolipoprotein-A2, HDL$_4$ phospholipids, HDL$_3$ apolipoprotein-A2, HDL$_4$ apolipoprotein-A2, HDL$_3$ free cholesterol, VLDL$_4$ cholesterol, VLDL$_4$ free cholesterol, VLDL$_2$ cholesterol to discriminate between female aripiprazole responders and non-responders at week 16 (AUC = 88.5%).

These associations are likely to be independent of variations in drug metabolism since the concentrations of these metabolites were not correlated with the circulating norescitalopram/escitalopram ratio. Collectively, the panel of lipoproteins comprising apolipoproteins A1 and A2, HDL-apolipoproteins A1 and A2,

HDL$_3$-free cholesterol, HDL$_3$-apolipoprotein A1, and HDL$_{2-3}$-apolipoprotein A2 showed a moderate ability (72.4% accuracy) to discriminate male escitalopram responders from non-responders.

In females, apolipoprotein A2, the apolipoprotein A2 component of HDL, and the apolipoprotein A2 and free cholesterol

components of HDL$_3$ were predictive of response to aripiprazole augmentation independent of age, BMI, and inflammation. At baseline, prior to escitalopram and aripiprazole treatment, LDL$_5$ subfractions were significantly higher in responders to the combination of escitalopram and aripiprazole compared to the non-responders. These features were also positively correlated with the dehydroaripiprazole/aripiprazole ratio at week 16, suggesting that individuals with inherently higher circulating amounts of LDL$_5$ subfractions experience a greater breakdown of aripiprazole to its major active metabolite dehydroaripiprazole. Both the parent drug and its active metabolite are partial agonists at dopamine D$_2$ receptors and in this study, the dehydroaripiprazole/aripiprazole ratio was found to be positively associated with treatment response. At present it is unclear if LDL$_5$ can influence the distribution or metabolism of aripiprazole, however, lipophilic drugs like escitalopram ($\log P = 3.8$) and aripiprazole ($\log P = 4.9$) bind to lipoproteins with moderate affinity[37]. Drug–lipoprotein binding alters the pharmacokinetic properties (e.g. distribution and clearance) of the drug and affects concentration–effect relationships by promoting drug uptake into tissues expressing specific lipoprotein receptors (30). Overall, a baseline lipoprotein signature comprising 10 HDL- and VLDL-related components was found to accurately predict (88.5% accuracy) female responders to aripiprazole augmentation.

With the current dataset, it was not possible to isolate the effect of food intake on the associations found. Future work should take dietary intake and fasting status into account. The lack of an independent validation dataset is also a limitation preventing these candidate biomarkers from being validated. Future research should explore this, controlling for other factors that can affect lipid status, such as smoking, menopausal status, and medication use (including hormonal medication). This is, to the best of our knowledge, the first time that individual lipoprotein subclasses (of varying size and density) have been investigated for their ability to predict antidepressant response, allowing us to identify associations that may have been masked in previous studies. Several methods can be used for lipoprotein subclass analysis including density-gradient ultracentrifugation, gel electrophoresis, ion mobility analysis, and NMR spectroscopy. Refinement of these platforms could lead to the development of a cost-effective, easy-to-use test capable of rapidly evaluating an individual's potential to respond to an antidepressant in the clinic. One example is an FDA-approved polyacrylamide tube gel electrophoresis system that is able to separate and quantify seven subclasses of LDL for clinical use[38].

This exploratory, hypothesis-generating study revealed the existence of a sex-dependent plasma signature of depression and treatment response to two commonly used pharmacological strategies. In this study, urinary markers were not identified, suggesting that blood may be more informative than urine for identifying metabolic predictors of response to escitalopram and aripiprazole.

Based on the identified markers, accurate predictions were made for treatment responses to escitalopram (72.4% accuracy) and aripiprazole (88.5% accuracy). In 2017, an estimated 26 million individuals were prescribed escitalopram, and 7 million were prescribed aripiprazole, with response rates of 66% and 61%, respectively[28,39]. Therefore, accurate prediction of treatment responses has the potential to impact the quality of life of millions of people and have wide economic benefits to healthcare systems. The integration of these predictive signatures with other markers identified via different modalities by the CAN-BIND network will facilitate the construction of a multi-level, personalized signature of MDD and of antidepressant response useful for clinical practice. Efforts to optimize the treatment pipeline are essential to counter the pressing, growing burden of MDD toward accurate personalized treatments.

## Methods

**Participants**. A total of 323 (204 females and 119 males) MDD and HC participants were recruited from six outpatient centers across Canada between August 2013 and December 2016[26–28]: Vancouver (Djavad Mowafaghian Centre for Brain Health), Calgary (Hotchkiss Brain Institute), Toronto (2 sites: University Health Network and Centre for Addiction and Mental Health), Hamilton (St. Joseph's Healthcare Hamilton), and Kingston (Providence Care, Mental Health Services). Recruitment draws upon outpatient-referral networks, community-based advertising, and dedicated knowledge translation activities. Participants were enrolled based on meeting eligibility criteria for either group. Individuals with no history of MDD or other psychiatric disorders were classified as healthy, while individuals with a MADRS score of ≥24 (indicating the presence of moderate-to-severe depressive symptoms) and a diagnosis of MDD confirmed with the Mini-International Neuropsychiatric Interview version 6.1, in a current major depressive episode of 3 months or longer were classified as 'depressed'. A complete list of exclusion and inclusion criteria can be found in[27]. Medication use was included as an exclusion criterion. Using these eligibility criteria, 112 HC participants (MADRS = 0.8 ± 1.7) and 211 MDD participants (MADRS = 29.9 ± 5.6) were identified. Participants provided written informed consent, and the study was conducted in compliance with the principles of Good Clinical Practice and relevant institutional research ethics boards. Blood and urine samples were collected as part of the Canadian Biomarker Integration Network in Depression Study-1 (CAN-BIND-1), a multi-center discovery study designed to identify predictors of MDD treatment response (ClinicalTrials.gov identifier: NCT01655706)[27,28].

**Study design**. The total duration of the study was 16 weeks. In phase I (baseline-week 8), the MDD group received escitalopram (an SSRI with a well-established antidepressant profile) at a dose of 10–20 mg daily. At the end of phase I (week 8), subjects were classified as 'responders' to escitalopram monotherapy if they achieved a 50% or greater reduction in MADRS score relative to baseline. In phase II (week 9-week 16), responders were maintained on the same escitalopram dose, while non-responders received escitalopram augmented with aripiprazole (2–10 mg). Response to the combination of escitalopram and adjunctive aripiprazole was defined as a 50% or higher reduction in MADRS score from baseline to week 16. A placebo group was not included in this study design as the focus of this study was to identify biomolecular predictors of treatment response (e.g. baseline versus week 8; baseline versus week 16) rather than the metabolic changes induced by escitalopram and aripiprazole (e.g. placebo group versus escitalopram group at week 8). Urine and blood samples were collected at baseline, week 2, week 8 and week 16 from both HC and MDD groups (Fig. 1) and matched with MADRS scores from the same visits. MADRS scores and plasma CRP concentrations (mean ± s.d.) of responders and non-responders to the two antidepressant drugs are reported in Supplementary Table 1 for each individual timepoint and sex. In addition, concentrations of escitalopram and aripiprazole, and their respective breakdown products norescitalopram and dehydroaripiprazole were measured in plasma by the Clinical Laboratory and Diagnostic Services at CAMH (Toronto, ON) at week 2, week 10 and week 16. Standard operating procedures (SOPs) were in place to ensure maximal consistency and reproducibility across recruitment sites, and sample collection protocols have been published in[27]. All data produced in this study are available from Brain-CODE (https://www.braincode.ca/)[40].

**Metabolic phenotyping**. Plasma and urine samples were analyzed by $^1$H NMR spectroscopy at the National Phenome Centre, Imperial College London. All steps of the analysis, from sample preparation and NMR data acquisition, were performed according to published protocols[41]. Upon collection, urine samples were centrifuged at 1200×$g$ at 4 °C for 5 min, and blood samples were centrifuged at 1600×$g$ for 15 min. The supernatant was collected and stored at −80 °C. Prior to analysis, urine samples were centrifuged at 12,000×$g$ at 4 °C for 5 min, and 540 μL of the supernatant was combined with 60 μL of potassium phosphate buffer containing the internal standard trimethylsilylpropionate (TSP). Samples were vortexed to mix and transferred to 5 mm NMR tubes. Similarly, plasma samples were centrifuged at 12,000×$g$ at 4 °C for 5 min. Plasma supernatants (350 μL) were combined with 350 μL of disodium phosphate buffer containing TSP, vortexed, and then transferred to 5 mm NMR tubes. To monitor technical variation potentially introduced at the sample preparation stage or due to the stability of the analytical approach, quality control (QC) samples were measured with the study samples. QC samples were prepared by pooling equal parts of each study sample and analyzed every 40 samples. Experiments were performed on a Bruker Advance III HD 600 MHz spectrometer operating at 14.1 T. Three separate experiments were performed on each plasma sample: standard 1D spectra using the 1D-NOESY presat pulse sequence, relaxation edited spin-echo using the 1D-Carr-Purcell-Meiboom-Gill (CPMG) presat pulse sequence and pseudo-2D spectra using a $J$-resolved sequence. For each urine sample, 1-D-NOESY presat and $J$-resolved pulse sequences were run. Automated processing of the spectra was performed using TopSpin 3.6 (Bruker Corporation, Germany) including spectral calibration, phase and baseline correction. The Bruker B.I.-LISA method (Bruker BioSpin 08/2019 T165319) was used to quantify lipoprotein main class (very low-density lipoprotein [VLDL], intermediate-density lipoprotein [IDL], low-density lipoprotein [LDL], and high-density lipoprotein [HDL]), their particle size and density (referred to as "subclass": VLDL$_{1-6}$, LDL$_{1-6}$, HDL$_{1-4}$ sorted according to increasing density and decreasing

size), and their building components (referred to as "subfraction": apolipoprotein, cholesterol, free cholesterol, phospholipids, triglycerides) in the plasma samples directly from their NMR spectra[42]. A list of all 112 lipoproteins measured can be found in Supplementary Data 1. For the urine samples, the B.I.QUANT-UR method was used (Bruker BioSpin 08/2019 T165319) to quantify 50 known compounds from their NMR spectra (Supplementary Data 2). PCA models were built on the urinary metabolic profiles and the plasma lipoprotein profiles of all study samples and the QC samples. The scores plots from these models showed that the profiles of the QC samples clustered together at the center of the plots (Supplementary Fig. 4) confirming low intra-study variation in these samples. This demonstrates the reliability and reproducibility of this analytical approach. The high reproducibility of the B.I.-LISA method for quantifying lipoprotein sub-classes has been previously shown[42], supporting its potential applicability for clinical purposes.

Following the acquisition of the urine profiles, the data was normalized using a probabilistic quotient approach to account for variability in sample dilutions. The urine variables were then log-transformed to account for a left-skewed distribution. As plasma is homeostatically controlled no normalization steps were applied to the lipoprotein profiles. No data imputation was necessary as there were no missing values. Metabolic profiles were filtered to remove variables with zero variance across the samples. Hierarchical clustering, based on Euclidean distances and the average linkage method, was used to identify samples that were outliers.

To quantify serum concentrations of escitalopram and aripiprazole and of their respective major metabolites, a mix of deuterated internal standards (Cerilliant) was added to 100 μL of each serum specimen, calibrator (Cerilliant) and QC (MassCheck Antidepressants/Neuroleptics, Level 1 and Level 2). Proteins were precipitated with 300 μL of 9:1 acetonitrile:methanol (Sigma-Aldrich) followed by 5 min centrifugation at 9000 rpm. 25 μL of each supernatant was diluted with 200 μL of 0.1% formic acid (Sigma-Aldrich) and then analyzed on the LC–MS/MS platform consisting of ThermoFisher TSQ Quantum Ultra mass spectrometer coupled with ThermoFisher Surveyor LC pump and HTC PAL autosampler fitted with a Kinetex F5 2.6 μm, 100 × 2.1 mm column (Phenomenex). Collision energies ranged from 14 to 20. Mass transitions (M + H) were monitored in SIM mode as follows: aripiprazole (448 → 285); dehydroaripiprazole (446 → 285); escitalopram (325 → 109); desmethyl-escitalopram (314 → 109) Quantification was performed against a calibration curve ranging from 10 to 1000 ng/mL for each analyte.

**Statistics and reproducibility**. A schematic representation of the analytical workflow is shown in Fig. 1. Statistical analysis was carried out in R (version 4.0.1). Weighted correlation network analysis was performed with the R package WGCNA (version 1.69)[43]. While originally developed for the study of gene co-expression analysis, the suitability of WGCNA for the analysis of metabolomic data has been demonstrated[44]. Correlation networks allow for system-level investigations of the metabolites (i.e. nodes) and their relationships (i.e. edges) in complex biological systems. They represent a powerful approach for analyzing multicol-linear data while preserving the information about the correlation structure of the data. For the current WGCNA analysis, the metabolites' correlation matrix was converted to a co-expression similarity matrix, calculated as the absolute value of the (Pearson) correlation coefficient between variable pairs (unsigned network). An unsigned network was chosen to allow for the clustering of metabolites with both positive and negative associations. The adjacency matrix was obtained by raising the similarity measure to the power $\beta$, where $\beta$ is the soft threshold (weighted networks). The value of $\beta$ was chosen by taking into account mean connectivity and the scale-free topology criterion, that is, the value for which the network approximates to a scale-free topology. Values of $\beta$ were adjusted for each model and referred to when reporting the results of the WGCNA analysis[45]. The final network was constructed on the topological overlap measure (TOM), computed as the normalized matrix product of the adjacency matrix with itself. Clusters of highly connected metabolites, referred to as "modules", were identified by average-linkage hierarchical clustering[45], with the dendrogram cut height for module merging (mergeCutHeight) set at 0.25. The deep split was set to 2, and the minimum cluster size was set to 3 for lipoproteins and 2 for urinary metabolites. The significance of the module eigengene (or "eigenmetabolite") with the outcome was calculated by univariate regression. Weighted correlation networks were constructed to identify clusters of highly connected metabolites explaining the variability in (i) MADRS score at baseline, (ii) the percentage MADRS reduction during phase I (between baseline and week 8), phase II (between week 8 and week 16) and during the entire trial (between baseline and week 16), as well as in the covariates of interest (i.e. age, BMI and CRP). Percentage change in MADRS score was chosen over absolute change to obtain a measure of improvement in depressive symptoms independent of the initial severity at baseline. Networks were built on all metabolic data from the entire cohort, as well as separately for males and females, to account for the known sex differences in metabolic signatures of depression and antidepressant response. Metabolites belonging to clusters exhibiting a significant association ($p < 0.05$) with the clinical variables were evaluated by a two-sided Mann–Whitney $U$ test to identify features discriminating MDD from HC groups or treatment responders from non-responders. The $p$ values of the WGCNA modules and of the Mann–Whitney $U$ test comparisons were corrected for multiple comparisons using the Benjamini–Hochberg correction with a false discovery rate

(FDR) of 20%. Only the metabolites that passed the FDR correction for multiple comparisons are reported. For these metabolites, two-tailed logistic regression models were built against a binary output (i.e. MDD vs. HC or responders vs. non-responders) and the receiver operating characteristic (ROC) curves were generated to evaluate the predictive ability of the selected metabolite panels. The 95% confidence intervals (CI) of the resulting area under the curve (AUC) were calculated using the DeLong method. A two-tailed partial correlation against MADRS was also performed to correct for the confounding effect of age, BMI and CRP. Additionally, for metabolites predictive of antidepressant response, Spearman correlation was carried out to investigate the relation between the baseline concentrations of the metabolites and the circulating levels of the antidepressant drugs, calculated as the ratio between norescitalopram/escitalopram and dehydroar-ipiprazole/aripiprazole. Norescitalopram and dehydroaripiprazole are the meta-bolic products of escitalopram and aripiprazole, respectively.

**Reporting summary**. Further information on research design is available in the Nature Research Reporting Summary linked to this article.

## Data availability

The data produced and used in this study are supported by The Canadian Biomarker Integration Network in Depression (CAN-BIND, https://canbind.ca/), which is an Integrative Discovery Program funded by the Ontario Brain Institute (OBI, https://braininstitute.ca/). In accordance with the Research Activity Agreements between CAN-BIND and OBI, data produced in this study must be submitted to Brain-CODE (https://www.braincode.ca/), an informatic platform created by the OBI to facilitate open-access of data generated from research funded by the OBI. Researchers requesting Data will provide OBI with written documentation of the proposed use of the Data in the form of an Research Ethics Board (REB) approval package which includes the full REB submission package and REB approval letter from their local Institutional REB. Should the REB determine that ethics review is not required, an exemption letter from the REB will be required. If the External Researcher is from an Institution that does not have a local REB, OBI will work with the External Researcher to identify a mutually acceptable REB for review. All documents not in English or French will require a certified translation copy. For detailed data access policy and procedure, please refer https://www.braininstitute.ca/research-data-sharing/brain-code or contact help@braincode.ca. Otherwise, data points underlying Figs. 2b, 3b, and 4b are presented in Supplementary Data 5–7.

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

## Acknowledgements

CAN-BIND is an Integrated Discovery Program carried out in partnership with, and financial support from, the Ontario Brain Institute, an independent non-profit corporation, funded partially by the Ontario government. The opinions, results and conclusions are those of the authors and no endorsement by the Ontario Brain Institute is intended or should be inferred. Additional funding is provided by the Canadian Institutes of Health Research (CIHR), Lundbeck, Bristol-Myers Squibb, Pfizer, and Servier. Funding and/or in kind support is also provided by the investigators' universities and academic institutions. All study medications are independently purchased at wholesale market values. This work was also supported by the Medical Research Council and National Institute for Health Research [grant number MC_PC_12025] and infrastructure support was provided by the National Institute for Health Research (NIHR) Imperial Biomedical Research Centre (BRC). G.C. is supported by the MRC (grant number MR/N014103/1).

## Author contributions

G.C. performed the analysis and drafted the manuscript under the direct supervision of J.A.F. and J.R.S. B.N.F., R.W.L., G.M.M., R.V.M., S.H.K., S.R., D.J.M., G.T., J.A.F. are the clinical site leads and/or principal investigators responsible for the conception and design of the study. They were involved in participant recruitment and data collection. All authors participated in the discussion regarding the interpretation of the data and provided comments on the manuscript.

## Competing interests

R.W.L. has received honoraria for ad hoc speaking or advising/consulting, or received research funds, from: Allergan, Asia-Pacific Economic Cooperation, BC Leading Edge Foundation, Canadian Institutes of Health Research, Canadian Network for Mood and Anxiety Treatments, Canadian Psychiatric Association, Hansoh, Healthy Minds Canada, Janssen, Lundbeck, Lundbeck Institute, Michael Smith Foundation for Health Research, MITACS, Myriad Neuroscience, Ontario Brain Institute, Otsuka, Pfizer, St. Jude Medical, University Health Network Foundation, and VGH-UBCH Foundation. R.V.M. has received consulting and speaking honoraria from Allergan, Janssen, KYE, Lundbeck, Otsuka, and Sunovion, and research grants from CAN-BIND, CIHR, Janssen, Lallemand, Lundbeck, Nubiyota, OBI and OMHF. B.N.F. received a research grant from Pfizer. SHK has received research funding or honoraria from the following sources: Abbott, Alkermes, Allergan, Boehringer-Ingelheim, Brain Canada, Canadian Institutes of Health Research (CIHR), Janssen, Lundbeck, Lundbeck Institute, Ontario Mental Health Foundation (OMHF), Ontario Brain Institute, Ontario Research Fund (ORF), Otsuka, Pfizer, Sanofi, Servier, St. Jude Medical, Sun Pharmaceuticals, and Sunovion. The remaining authors declare that there is no competing interests.
