## [Peer Review File · Communications Biology]

Reviewers' comments:

Reviewer #1 (Remarks to the Author):

Caspani et al. describe a study in which they aim to identify metabolomics signatures associated with depression and predictors of antidepressant response. This study is highly relevant and features a number of important and partially unique strengths: relatively large sample size, inclusion of men and women, repeated samples, plasma and urine metabolomics. The manuscript is also written well. However, the analysis workflow is complicated and includes several different methods and outcomes or versions of the same outcome. In its present form, the study leaves the reader wondering if the presented workflow is the only sequence of steps that results in the identification of statistically significant associations. Another worrisome part relates to the urine metabolomics. No metabolite changes were identified using urine samples. Have the authors performed a careful quality control analysis? Furthermore, there are a number of additional issues that need to be addressed:

Were there any QC samples measured to assess the technical reproducibility of the measurement platform on these samples?

Were all samples collected using the same protocol? This is of particular relevance in the current study which involved multiple study sites. Did the authors perform analyses to identify any potential study-site specific effects?

Hierarchical clustering was used to remove outliers. It is not clear if this statement refers to metabolites or samples. This step should be described in more detail.

Fasting has a large impact on plasma metabolites. How did the analysis account for this sample collection characteristic?

Among women, menopausal status and hormone use lead to changes in metabolite levels. How did the analysis account for these factors?

Additional factors are known to impact metabolite levels: disease, medication use, smoking status, etc. How did the authors take these into account?

This study features a complex and potentially too complicated analysis workflow. Authors should clarify the reasons for choosing each step of the analysis. For example, why did the authors decide to use WGCNA for step 1 and not linear regression (or conditional logistic regression, depending on the outcome), adjusting for any potential confounders? It is difficult to understand why any true associations would not be captured by this much simpler, easy to understand approach that represents the standard analysis method for these types of studies. While WGCNA takes into account the correlation structure among metabolites, its results are difficult to interpret, especially as metabolite modules are not consistent across analyses. There are multiple other approaches to truly account for correlations when adjusting for testing multiple hypotheses (e.g., step down min P approach by Westfall and Young). Using regression in step 1 would also mean that the Mann-Whitney U test would not be needed anymore, considerably simplifying the analysis. It is also difficult to understand why sometimes a continuous and sometimes a binary outcome is used. Another example would be the use of penalized regressions (e.g., LASSO, elastic net) for the prediction part. These methods were developed with the goal of identifying the most predictive group of metabolites in the presence of correlations.

The p-values for WGCNA modules and their associations with the outcome should be corrected for testing multiple hypotheses.

In order to better quantify the prediction capability of the identified group of metabolites, the authors

should report area under the ROC curve (AUC) and corresponding CI.

Reviewer #3 (Remarks to the Author):

Review of manuscript #COMMSBIO-20-3075-T

This manuscript describes the changes in lipids associated with major depressive disorder in men and women compared to healthy controls and also examines the lipids as predictors of treatment response. Overall, the authors did a good job in presenting complex analysis and results in a well-organized and easy-to-read fashion.

Quality of the work: This work is typical in terms of a metabolomics paper where tissue samples, typically serum/plasma &/or urine from patients and healthy controls are profiled retrospectively and the altered metabolites are presented as signatures of the disease severity or predictors of prognosis or treatment response. There are no replications in another dataset or functional interpretations of the metabolite changes presented. That being said I would also like to mention that these findings are important to be published as 1) the metabolites mentioned to be perturbed have not been presented before in such detail, 2) we need to understand more about MDD pathophysiology as this disease has reached epidemic proportions in recent times and 3) even with the limitations, the authors have done a good job in presenting their work.

Major points of critique:

1) A major item in metabolomics data analysis is data preprocessing. How the data was preprocessed, normalized, missing value imputation done or not (if done, how was it done), whether the data was transformed or not, outlier detection etc all should be mentioned in detail.

2) The WGCNA procedure should be mentioned with more details in terms of the parameters used for the final networks, e.g., signed or unsigned networks, what correlation method was used to calculate the adjacency matrix, deep split, minimum cluster size etc should all be mentioned. These are important for 1) method reproducibility and 2) for future data analysts adopting similar methods for their own datasets.

3) In partial correlations or logistic regression models, the authors adjusted for covariates age, BMI and CRP. What about the variability due to different patient recruitment sites?

4) Also, the values for each of the variables should be presented more clearly. For example, since analysis were presented pre and post treatment, we would like to know the MADRS values (mean +- SD) pre and post treatment and in responders versus non-responders. We would like to know the CRP values pre and post treatment and in responders vs non-responders.

5) A table containing the metabolites and their pre and post treatment values or log fold changes should be presented along with their p-values from the univariate analysis done. From the results we understand which lipid molecules were up or down at baseline or with treatment but we have no idea about by how much they were increased or decreased.

6) Were the patients fasted or non-fasted before blood draw? Lipids are greatly impacted by fasting too besides diet. A more detailed section on limitations of the study should be presented. For example, the authors mention about predictive metabolite markers of response but predictive analysis rigor is not presented, no training-testing data sets, cross-validation or other external validation or replication data presented in the analysis. It should be clearly mentioned in the limitation section of the study that these markers may have potential as predictive metabolite markers and should be tested in a larger independent data sets.

7) Did any of the metabolites that were higher or lower in MDD patients at baseline change with changes in MADRS scores post treatment?

8) The metabolites whose baseline levels were significantly higher or lower in patients who eventually responded to treatment versus who did not, were they significant irrespective of baseline depression severity? This is important to know if they are thought to be potential predictors of treatment response.

Below we provide a detailed response to each point raised by the reviewers.

Reviewer #1 (Remarks to the Author):

1) In its present form, the study leaves the reader wondering if the presented workflow is the only sequence of steps that results in the identification of statistically significant associations. Another worrisome part relates to the urine metabolomics. No metabolite changes were identified using urine samples. Have the authors performed a careful quality control analysis?

Careful quality control analysis was performed. We have added this information to the methods section (**Page 7; lines 115-119**) and have added a figure to the Supplementary Information showing these results (Supplementary Figure S2):

To monitor technical variation potentially introduced at the sample preparation stage or due to the stability of the analytical approach, quality control (QC) samples were measured with the study samples. QC samples were prepared by pooling equal parts of each study sample and analyzed after every 40 study samples.

*PCA models were built on the urinary metabolic profiles and the plasma lipoprotein profiles of all study samples and QC samples. The scores plots from these models showed that the profiles of the QC samples clustered together at the center of the plots (**Supplementary Figure S2**) confirming low intra-study variation in these samples. This demonstrates the reliability and reproducibility of this analytical approach.*

Supplementary Figure S2. PCA scores plots of study samples and quality control samples. PCA models constructed on the A) lipoprotein profiles and B) urinary metabolic profiles of all study samples (orange) and the pooled quality control (QC) samples (blue).

We do not believe that absence of significant findings in the urine data is worrisome. The urinary metabolic profiles studied in this analysis comprised 50 quantified metabolites. This relatively narrow view of the metabolome combined with the heterogeneous nature (age, sex, SES, BMI) of the CAN-BIND study population may explain the lack of features associated with MDD/MADRS and treatment response. To clarify this limitation we have added the following sentences to the discussion (**page 17; lines 364-368**):

However, it should be noted that the urinary metabolome assessed here comprised 50 metabolites and therefore represented a relatively narrow snapshot of the urinary metabolome. The broader metabolomic view

afforded by the application of untargeted approaches may yield biochemical signatures in the urine with potential to predict antidepressant treatment responses.

2) Were there any QC samples measured to assess the technical reproducibility of the measurement platform on these samples?

Yes – as stated above. Details on the quality control steps taken to ensure reproducibility are now described on **page 7; lines 116-119** and **lines 133-139**). In brief, the reproducibility of our findings was ensured by the following precautions:

- (1) NMR analysis was performed by the MRC-NIHR National Phenome Centre (NPC) at Imperial College London (UK). The NPC is a world-leading centre for metabolic phenotyping, which develops metabolic profiling methods and protocols for research and diagnostics.
- (2) The NPC uses strict, published protocols to ensure reproducibility. These are cited in the manuscript on **page 7; lines 115-116**).
- (3) Along with the study samples, each NMR run also included quality control (QC) samples, which allow the total variation across the preparation and analysis stages (*i.e.* intra-study variation) to be monitored. In addition, long term reference samples were analysed within the run. This is a large pool of biologically relevant specimens independent of the study, that allow the analytical variation over longer periods of time (*i.e.* inter-study variation) to be assessed. This can monitor aspects such as changes in instrument performance. We have included the PCA scores plots of the QC samples and study samples (**Supplementary Figure S2**) to show the validity of our data.
- (4) The targeted lipoprotein subclass analysis performed in this study has been demonstrated to be highly reliable and reproducible and suitable for clinical purposes. The reference to the Jiménez *et al.* paper (2018) has been added to the revised manuscript on **page 7; lines 137-139**) to provide the reader with this information.

3) Were all samples collected using the same protocol? This is of particular relevance in the current study which involved multiple study sites. Did the authors perform analyses to identify any potential study-site specific effects?

Yes. Standard operating procedures (SOPs) were in place for all steps of the study, including sample collection, preparation, and NMR data acquisition. As a result, samples were collected following the same protocol across recruitment sites. We have added this information on **pages 6-7; lines 110-113** of the manuscript, including a reference to the Lam *et al.* paper (2016), which has a detailed description of the protocol used for sample collection in the CAN-BIND study.

Standard operating procedures (SOPs) were in place to ensure maximal consistency and reproducibility across recruitment sites, and sample collection protocols have been published in ²⁷.

In addition, PCA analysis was performed to assess whether study site was a source of metabolic variation. The scores plots from these models have been added to the supplementary material as **Supplementary Figure S4**. These plots show overlap in the metabolic profiles from the different study sites indicating that site did not introduce metabolic variation. The following text has been added to the manuscript (**page 9; line 193-196**):

PCA models were constructed on the urine metabolic profiles and plasma lipoprotein profiles to assess if study site was source of variation. From the scores plots of these models (**Supplementary Figure S4**) no clustering was apparent based on the site that the samples were collected from.

Supplementary Figure S4. PCA plots of lipoprotein and urinary profiles colored by recruitment site. The lipoprotein (A) and urinary (B) metabolomes of the participants were not affected by study site, as shown by the largely overlapping distributions in the scores plots. CAM = Centre for Addiction and Mental Health; MCU = McMaster University; QNS = Queen's University; TGH = Toronto General/Western Hospital; UBC = University of British Columbia; UCA = University of Calgary.

4) Hierarchical clustering was used to remove outliers. It is not clear if this statement refers to metabolites or samples. This step should be described in more detail.

Hierarchical clustering was performed to identify observations (*i.e.* participants) with a distinct metabolic profile. This approach relies on the fact that the height of the dendrogram in a given branch is proportional to the distance (or dissimilarity) between observations. More information on this step of the analysis can now be found on **page 8 (lines 144-145)**:

Metabolic profiles were filtered to remove variables with zero-variance across the samples. Hierarchical clustering, based on Euclidean distances and the average linkage method, was used to identify samples that were outliers.

5) Fasting has a large impact on plasma metabolites. How did the analysis account for this sample collection characteristic?

Blood was collected from non-fasted participants. As fasting status was variable across all participants (healthy controls and MDD; responders and non-responders), it is unlikely to be contributing to the observed results. Moreover, recent investigations even suggest that lipoprotein measurements may be more accurate in non-fasting samples. We refer the reviewers to the Nordestgaard *et al.* (2016) paper, which argues that the difference between random non-fasting and fasting lipoproteins is not clinically significant.

Nordestgaard, B.G., Langsted, A., Mora, S., Kolovou, G., Baum, H., Bruckert, E., et al. (2016). Fasting is not routinely required for determination of a lipid profile: Clinical and laboratory implications including flagging at desirable concentration cut-points - A joint

consensus statement from the European Atherosclerosis Society and European Fed. Eur. Heart J. 37: 1944–1958.

In addition, the fact that we detect these biomarkers of future treatment response against a background of variable fasting states demonstrates promise to translate these findings into the clinic. Nevertheless, knowledge of the participant's diet and fasting state can be beneficial for the interpretation of the results. For this reason, we state the lack of this information as a potential limitation on **page 16; lines 349-353**, and we encourage future validation studies to take fasting and diet into account.

With the current dataset, it was not possible to isolate the effect of food intake on the associations found. Future work should take dietary intake and fasting status into account.

6) Among women, menopausal status and hormone use lead to changes in metabolite levels. How did the analysis account for these factors?

It was not possible to account for menopausal status and hormone use, as this data was not collected during the study. We discuss this limitation on **page 16; lines 350-353**.

The lack of an independent validation dataset is also a limitation preventing these candidate biomarkers from being validated. Future research should explore this, controlling for other factors that can affect lipid status, such as smoking, menopausal status, and medication use (including hormonal medication).

However, the largest sources of variation in hormone fluctuations are sex and age, both of which were controlled for in the analysis. Additionally, our data can be generalized to the population independently of menopausal status and hormone use.

7) Additional factors are known to impact metabolite levels: disease, medication use, smoking status, etc. How did the authors take these into account?

We have no information on whether the participants were smokers. As stated above, we now mention this limitation on **page 16; line 352**. With regards to disease, the exclusion criteria imply that individuals with significant medical history were not recruited on the study. Participants were also free from psychotropic medication and substance abuse. The following sentence has been added to the methods section to emphasize this (**Page 6; line 92**):

Psychotropic medication use and substance abuse was included as exclusion criteria.

8) This study features a complex and potentially too complicated analysis workflow. Authors should clarify the reasons for choosing each step of the analysis. For example, why did the authors decide to use WGCNA for step 1 and not linear regression (or conditional logistic regression, depending on the outcome), adjusting for any potential confounders? It is difficult to understand why any true associations would not be captured by this much simpler, easy to understand approach that represents the standard analysis method for these types of studies.

While the points raised by reviewer #1 are correct, we argue that WGCNA is the most suitable approach for our data and aims. Network analysis performed by WGCNA provides a system-level insight into the baseline biochemical profile of MDD patients that respond to antidepressant treatment as opposed to those that do not respond. An additional advantage of correlation network approaches is that, by clustering metabolites based on biologically-

relevant correlated expression profiles, they are sensitive to metabolites with low abundance and/or small fold changes between groups. A large number of papers applying WGCNA to metabolomic data have been published in the last few years (Carmelo et al., 2020; Hsu et al., 2020; Vernocchi et al., 2020, to name a few), supporting the suitability of correlation network approaches for metabolic data. As requested by the reviewer, we have added text to the methods section to clarify why we chose this approach (**Pages 8-9; lines 158-175**).

Reviewer #1 suggests that linear regression is a simpler, yet as-effective approach, than WGCNA. We argue that linear regression is not appropriate in our analysis due to (1) the high dimensionality of the data, and (2) the presence of highly correlated variables (*i.e.* multicollinearity). It is known that multicollinearity can lead to overfitting in multiple linear regression, and for this reason it was excluded.

Another example would be the use of penalized regressions (e.g., LASSO, elastic net) for the prediction part. These methods were developed with the goal of identifying the most predictive group of metabolites in the presence of correlations.

As suggested by Reviewer #1, sparse methods like Lasso and Elastic Net regression can overcome the problem of multicollinearity by selecting some predictors while shrinking (the coefficient of) others to zero. However, this approach is not suitable for our data. In fact, all lipoproteins are regulated by the same biochemical processes and are intrinsically connected (*i.e.* VLDL is metabolized to IDL, which is metabolised into LDL, *etc.*), meaning that a feature selection method like Lasso, which arbitrarily chooses a specific variable over another highly-correlated one, can result in the loss of biologically-relevant information. Instead, WGCNA leverages the correlation structure of the lipoprotein data to obtain clusters of metabolites that can be easily studied against an external trait (e.g. treatment response). WGCNA maintains this “structural” information on the relationship amongst metabolites while relating each cluster with the outcome.

While WGCNA takes into account the correlation structure among metabolites, its results are difficult to interpret, especially as metabolite modules are not consistent across analyses.

The fact that the metabolite modules change across sexes is of interest, as it signifies that the relationship between lipoproteins is different in males vs females, and would have not been captured by sparse regression methods. The interpretation of the WGCNA can be facilitated by the identification of the module membership, which is calculated as the signed correlation of each metabolite with their module eigengene. This value ranges from -1 to +1 and indicates how close each metabolite is to its respective module. We have added this information on “module membership”, the correlation between the metabolite and the module it belongs to, in the **Supplementary Table S5** and **S6** to facilitate the understanding of the WGCNA output.

Using regression in step 1 would also mean that the Mann-Whitney U test would not be needed anymore, considerably simplifying the analysis.

To improve the interpretability of our work to readers external to the metabolomic field, we also report the Mann-Whitney U test. Given that the current work spans the field of neuroscience and pharmacology, we found that this was an appropriate step to increase the accessibility of the work to readers of *Communications Biology*. It also provides a second layer of confidence when evaluating the significance of the lipoproteins in each module against the outcome.

It is also difficult to understand why sometimes a continuous and sometimes a binary outcome is used.

Finally, a continuous outcome (change in MADRS score) was used to perform WGCNA, as this approach is based on identifying the correlations between a module and a continuous outcome. Continuous outcomes provide higher resolution than binary outcomes and were preferred, when possible, in our workflow. A binary outcome (response vs no-response) was used to evaluate the predictive potential of the panel of lipoproteins identified via network analysis in a logistic regression, as this is the standard procedure for assessing a biomarker's diagnostic ability.

9) The p-values for WGCNA modules and their associations with the outcome should be corrected for testing multiple hypotheses.

We carried out the Benjamini-Hochberg correction with a false discovery rate (FDR) of 20% on all modules, and all remained significant after comparison with the critical threshold. This is stated in the method section on **page 9 (lines 185-187)**.

The p values for both the WGCNA modules and the Mann-Whitney U test comparisons were corrected for multiple comparisons using the Benjamini-Hochberg correction with a false discovery rate (FDR) of 20%.

10) In order to better quantify the prediction capability of the identified group of metabolites, the authors should report area under the ROC curve (AUC) and corresponding CI.

We report the AUC with the 95% confidence intervals for escitalopram (AUC = 72.4%, CI: 0.5974-0.8509) on **page 13; lines 257-258**, and for aripiprazole (AUC = 88.5%, CI: 0.7922-0.9776) on **page 14; lines 292-293**.

This is also mentioned in the methods on **page 9 (lines 191-192)**.

The 95% confidence intervals (CI) of the resulting area under the curve (AUC) were calculated using the DeLong method.

Reviewer #3 (Remarks to the Author):

1) A major item in metabolomics data analysis is data preprocessing. How the data was preprocessed, normalized, missing value imputation done or not (if done, how was it done), whether the data was transformed or not, outlier detection etc all should be mentioned in detail.

This information has been added to the methods section (**page 7; lines 124-125**):

Automated processing of the spectra was performed using TopSpin 3.6 (Bruker Corporation, Germany) including spectral calibration, phase and baseline correction.

And on **page 8 (lines 140-145)**:

Following the acquisition of the urine profiles, the data was normalized using a probabilistic quotient approach to account for variability in sample dilutions. The urine variables were then log-transformed to account for a left-skewed distribution. As plasma is homeostatically controlled no normalization steps were applied to the lipoprotein profiles. No data imputation was necessary as there were no missing values. Metabolic profiles were filtered to remove variables with zero-variance across the samples. Hierarchical clustering, based on Euclidean distances and the average linkage method, was used to identify samples that were outliers.

2) The WGCNA procedure should be mentioned with more details in terms of the parameters used for the final networks, e.g., signed or unsigned networks, what correlation method was used to calculate the adjacency matrix, deep split, minimum cluster size etc should all be mentioned. These are important for 1) method reproducibility and 2) for future data analysts adopting similar methods for their own datasets.

As stated for Reviewer 1 point 8, this information has been added on **page 8-9 (lines 158-175)**.

Weighted correlation network analysis was performed with the R package WGCNA (version 1.69)³². While originally developed for the study of gene co-expression analysis, the suitability of WGCNA for the analysis of metabolomic data has been demonstrated³³. Correlation networks allow for system-level investigations of the metabolites (i.e. nodes) and their relationships (i.e. edges) in complex biological systems. They represent a powerful approach for analyzing multicollinear data while preserving the information about the correlation structure of the data. For the current WGCNA analysis, the metabolites' correlation matrix was converted to a co-expression similarity matrix, calculated as the absolute value of the (Pearson) correlation coefficient between variable pairs (unsigned network). An unsigned network was chosen to allow for the clustering of metabolites with both positive and negative associations. The adjacency matrix was obtained by raising the similarity measure to the power β , where β is the soft threshold (weighted networks). The value of β was chosen by taking into account mean connectivity and the scale-free topology criterion, that is, the value for which the network approximates to a scale-free topology. Values of β were adjusted for each model, and referred to when reporting the results of the WGCNA analysis³⁴. The final network was constructed on the topological overlap measure (TOM), computed as the normalized matrix product of the adjacency matrix with itself. Clusters of highly-connected metabolites, referred to as "modules", were identified by average-linkage hierarchical clustering³⁴, with the dendrogram cut height for module merging (mergeCutHeight) set at 0.25. Deep split was set to 2, and the minimum cluster size was set to 3 for lipoproteins and 2 for urinary metabolites.

3) In partial correlations or logistic regression models, the authors adjusted for covariates age, BMI and CRP. What about the variability due to different patient recruitment sites?

See response to Reviewer 1, point 3.

4) Also, the values for each of the variables should be presented more clearly. For example, since analysis were presented pre and post treatment, we would like to know the MADRS values (mean +- SD) pre and post treatment and in responders versus non-responders. We would like to know the CRP values pre and post treatment and in responders vs non-responders.

Thank you for this suggestion. We have added **Supplementary Table S2** which summarises this information. The table is referred to on **page 6** of the manuscript (**lines 107-109**).

*MADRS scores and plasma CRP concentrations (mean \pm s.d.) of responders and non-responders to the two antidepressant drugs are reported in **Supplementary Table S2** for each individual timepoint and sex.*

5) A table containing the metabolites and their pre and post treatment values or log fold changes should be presented along with their p-values from the univariate analysis done. From the results we understand which lipid molecules were up or down at baseline or with treatment but we have no idea about by how much they were increased or decreased.

We have added **Supplementary Tables S5** and **S6** in the manuscript. **Supplementary Tables S5** shows the (log₂) fold-change of the ratio of the metabolite concentrations at week 8 over baseline (for escitalopram response). This was calculated for metabolites that belong

to those modules that were significantly associated with a reduction in MADRS in phase I (in males). The p values shown were obtained using the Mann-Whitney U test. **Supplementary Tables S6** shows the same for the metabolites that were associated with a reduction in MADRS during phase II (in females). In this case, the fold change was calculated from the ratio of the metabolite concentrations at week 16 over baseline (for aripiprazole response). These tables are referred to on **page 12 (lines 253-255)**:

*The pre- and post-treatment fold change of the metabolites belonging to the significant modules is shown in **Supplementary Table S4** for both male responders and non-responders.*

And **page 13; line 283-284** of the manuscript:

*Individual fold changes in these lipoprotein features from week 16 to baseline are reported in **Supplementary Table S5**.*

6) Were the patients fasted or non-fasted before blood draw? Lipids are greatly impacted by fasting too besides diet. A more detailed section on limitations of the study should be presented. For example, the authors mention about predictive metabolite markers of response but predictive analysis rigor is not presented, no training-testing data sets, cross-validation or other external validation or replication data presented in the analysis. It should be clearly mentioned in the limitation section of the study that these markers may have potential as predictive metabolite markers and should be tested in a larger independent data sets.

Participants were not fasted prior to blood collection (please see response to comment 5 by Reviewer #1). We have stated the limitations of our study on **page 16 (lines 349-350)**. We agree that the results of the present study need validation in a larger cohort, and we have stated this clearly on **page 16 (lines 350-352)**.

7) Did any of the metabolites that were higher or lower in MDD patients at baseline change with changes in MADRS scores post treatment?

When considering male and female participants together, LDL-6 and its subfractions (triglycerides, cholesterol, free cholesterol, phospholipids, apolipoprotein B) were found to be positively associated with MADRS score (*i.e.* higher in MDD than in healthy controls). These same lipoproteins were highlighted as positively correlated with MADRS reduction during phase II in female participants (module 5 in the WGCNA analysis, shown in panel C of **Supplementary Figure S4**), but were not significantly different between aripiprazole responders and non-responders at week 16 in the Mann-Whitney U test.

As for males, WGCNA highlighted a negative association between HDL-cholesterol, HDL2-cholesterol, LDL2, LDL2-cholesterol, LDL2-free cholesterol, LDL3-free cholesterol, LDL2-phospholipids, LDL2-apolipoprotein B and MADRS at baseline (*i.e.* lower in MDD than in healthy controls). While some of these lipoproteins overlapped with those highlighted by WGCNA constructed against MADRS change (see panel B of **Supplementary Figure S4**), none of them were included in the panel of biomarkers with predictive ability against escitalopram response.

8) The metabolites whose baseline levels were significantly higher or lower in patients who eventually responded to treatment versus who did not, were they significant irrespective of

baseline depression severity? This is important to know if they are thought to be potential predictors of treatment response.

Yes, and we have now stated this explicitly in the manuscript (**page 9, lines 176-181**):

Weighted correlation networks were constructed to identify clusters of highly connected metabolites explaining the variability in (i) MADRS score at baseline, (ii) the percentage MADRS reduction during phase I ($\Delta \text{MADRS}_{\text{week } 8} - \text{MADRS}_{\text{baseline}}$), and (iii) the percentage MADRS reduction during phase II ($\Delta \text{MADRS}_{\text{week } 16} - \text{MADRS}_{\text{week } 8}$), as well as covariates of interest (i.e. age, BMI and CRP). Percentage change in MADRS score was chosen over absolute change to obtain a measure of improvement in depressive symptoms independent of the initial severity at baseline.

Since metabolites were tested against the relative (as opposed to absolute) change in MADRS, the results reported in the manuscript are independent of the initial MADRS score at baseline. For example, a change in MADRS from 30 to 15 (= absolute change of 15 points on the MADRS scale) was equivalent to a MADRS change from 60 to 30 (= absolute change of 30), as both participants experience a 50% reduction. This means that the panel of biomarkers identified by our study are valid irrespective of initial depression severity.

We thank you again for considering publication of our manuscript and we look forward to hearing back from you.

Kind regards,

Giorgia Caspani, Gustavo Turecki, Raymond W. Lam, Roumen V. Milev, Benicio N. Frey, Glenda M. MacQueen, Daniel J. Müller, Susan Rotzinger, Sidney H. Kennedy, Jane A. Foster, Jonathan Swann

References

- Carmelo, V.A.O., Banerjee, P., Silva Diniz, W.J. da, and Kadarmideen, H.N. (2020). Metabolomic networks and pathways associated with feed efficiency and related-traits in Duroc and Landrace pigs. *Sci. Rep.* 10: 255.
- Hsu, M.J., Karkossa, I., Schäfer, I., Christ, M., Kühne, H., Schubert, K., et al. (2020). Mitochondrial transfer by human mesenchymal Stromal cells ameliorates hepatocyte lipid load in a mouse model of NASH. *Biomedicines* 8: 350.
- Jiménez, B., Holmes, E., Heude, C., Tolson, R.F., Harvey, N., Lodge, S.L., et al. (2018). Quantitative Lipoprotein Subclass and Low Molecular Weight Metabolite Analysis in Human Serum and Plasma by ¹H NMR Spectroscopy in a Multilaboratory Trial. *Anal. Chem.* 90: 11962–11971.
- Lam, R.W., Milev, R., Rotzinger, S., Andreazza, A.C., Blier, P., Brenner, C., et al. (2016). Discovering biomarkers for antidepressant response: Protocol from the Canadian biomarker integration network in depression (CAN-BIND) and clinical characteristics of the first patient cohort. *BMC Psychiatry* 16: 1–13.
- Nordestgaard, B.G., Langsted, A., Mora, S., Kolovou, G., Baum, H., Bruckert, E., et al. (2016). Fasting is not routinely required for determination of a lipid profile: Clinical and laboratory implications including flagging at desirable concentration cut-points - A joint

consensus statement from the European Atherosclerosis Society and European Fede. Eur. Heart J. 37: 1944–1958.

Vernocchi, P., Gili, T., Conte, F., Chierico, F. Del, Conta, G., Miccheli, A., et al. (2020). Network analysis of gut microbiome and metabolome to discover microbiota-linked biomarkers in patients affected by non-small cell lung cancer. Int. J. Mol. Sci. 21: 8730.

REVIEWERS' COMMENTS:

Reviewer #2: See attachment

Reviewer #3 (Remarks to the Author):

I have carefully read the rebuttal and the associated edits and additions the authors have made in the paper based on the suggestions/comments the reviewers had, including myself. They did an excellent job in their corrections and I am fully supportive of the publication of this manuscript.

I have also read their rebuttal to reviewer #1 and with my knowledge in metabolomics I am of the opinion that they have answered and taken needed actions on all the questions/critiques that this reviewer had.

This was a good manuscript to begin with and now it has become much better after all their edits and insertions.

Reviewer #4 (Remarks to the Author):

The authors have done a good job in addressing the critiques raised in the previous round of review. In my opinion, the manuscript can be accepted for publication as is.

Dear Prof Bahlo and reviewers,

Thank you for your positive feedback on our manuscript "Metabolomic signatures associated with depression and predictors of antidepressant response in humans: A CAN-BIND-1 report" (COMMSBIO-20-3075-T).

We have addressed the concerns raised by reviewer #2 and edited the manuscript to comply with the editorial requirements. Please find below our response to the remaining comments from Reviewer #2:

Overall, the manuscript is very well written, the methodology is excellent, and it provides important information for the field, particularly given the current focus on identifying biomarkers to improve treatment selection for depression. However, four major issues dampen the enthusiasm: Absence of validation in an independent data set, a more thorough assessment of the CVD risk factors in the population evaluated and lack of both a comparator in each phase and most importantly lack of a placebo arm as it relates to prediction of outcomes.

Thank you for your positive feedback on the quality of our work. We are aware that the absence of a validation cohort is a limitation of the current work and have stated this clearly in the discussion (**pages 12-13, lines 259-264**):

"With the current dataset, it was not possible to isolate the effect of food intake on the associations found. Future work should take dietary intake and fasting status into account. The lack of an independent validation dataset is also a limitation preventing these candidate biomarkers from being validated. Future research should explore this, controlling for other factors that can affect lipid status, such as smoking, menopausal status, and medication use (including hormonal medication)."

Regarding the issue of CVD risk raised by Reviewer #2, we are confident that this does not pose a threat to the validity of our results. Our findings remain significant when adjusting for BMI (obesity) and CRP (inflammation), two major risk factors for CVD.

We disagree that the lack of a placebo group is a major issue. This work sought to characterize the metabolic variation associated with depression (comparing metabolic profiles of MDD patients and healthy control individuals) and to identify biomarkers predictive of treatment response (variables in the baseline metabolic profiles that predict MADRS change following escitalopram/aripiprazole). A placebo group would allow us to study the metabolic impact of the treatments (metabolic profiles following treatment versus placebo) however, this comparison was not made and was not the focus of this report. Furthermore, the inclusion of a placebo group is not possible at this stage and would require an additional study. We do not believe that this substantial additional work would significantly improve this manuscript and is beyond the scope of this report. To clarify this point we have added the following sentence to the Methods section (**page 14; lines 311-314**):

"A placebo group was not included in this study design as the focus of this study was to identify biomolecular predictors of treatment response (e.g. baseline versus week 8; baseline versus week 16) rather than the metabolic changes induced by escitalopram and aripiprazole (e.g. placebo group versus escitalopram group at week 8)."

We believe that the relevance and novelty of these findings, especially, related to the importance of (i) studying sex differences and (ii) taking into account lipoprotein size and density, will guide future investigations into the role of plasma lipoproteins in predicting antidepressant response and merit publication in *Communications Biology*.

In response to the minor comments raised by Reviewer #2:

1. Throughout the results and discussion, the authors refer to response to aripiprazole. It should be made clear that this is actually response to combination escitalopram and aripiprazole in the female non-responders to phase 1 escitalopram. It is possible that the metabolic predictors identified have to do with the combination or the augmentation strategy, rather than strictly to aripiprazole itself.

We have clarified that aripiprazole was used as augmentation therapy throughout the manuscript.

2. While there is a supplementary table detailing the demographic and clinical characteristics of the sample, it would be helpful to include a brief paragraph at the start of the results to provide some information to the reader since not all readers will have access to the supplementary tables. Similarly, it would be helpful to include either a consort diagram of the flow and response rates of the participants, or at the very least include the response rate information in the text (both for phase 1 and phase 2). The paper reports n=83 were analyzed in phase 2, suggesting that the response rate in phase 1 was 60.7% (128/211). Is this correct?

In line with the suggestion of both Reviewer #2 and the editorial office, we have moved the table detailing the demographics and clinical characteristics of the CAN-BIND 1 cohort to the main manuscript (now **Table 1**). This table includes response rates to both escitalopram (47%) and aripiprazole (75%). The response rate in phase 1 is lower than 60.7% due to dropouts during the clinical trial. The number of dropouts and missing data are also shown in the demographics table. The following text was also added to the results (**page 6, lines 83-101**):

*“Clinical characteristics of the CAN-BIND 1 cohort. A total of 323 participants were recruited from 6 outpatient centers across Canada between August 2013 and December 2016²⁶⁻²⁸. These consisted of 112 healthy controls (HC) (Montgomery-Åsberg Depression Rating Scale [MADRS]=0.8±1.7) and 211 depressed (MDD) participants (MADRS=29.9±5.6). The groups were matched for age (N=323, Mann-Whitney U=17229.5, p=0.252) and sex (N=323, $\chi(1)=0.0041$; p=0.9491). Participants with clinical depression underwent a 16-week two-phase treatment trial: escitalopram (10-20 mg) was administered during phase I (baseline to week 8), and escitalopram either alone (in responders) or augmented with aripiprazole (2-10 mg, in non-responders) was administered during phase II (week 9 to week 16), as shown in **Figure 1**. Plasma and urine samples were collected pre-treatment and analyzed by ¹H nuclear magnetic resonance (NMR), giving rise to a panel of 112 lipoproteins (**Supplementary Data 1**) and 50 urinary metabolites (**Supplementary Data 2**). PCA models constructed on the urine metabolic profiles and plasma lipoprotein profiles of all participants showed that no clustering was apparent based on the site that the samples were collected from, demonstrating that the metabolic profile was not affected by study location (**Supplementary Figure 1**). Plasma data from 12 participants and urine data from 9 participants were excluded as they did not pass quality control checks. In addition, 31 participants dropped out of the study before termination of the clinical trial. The N given in the following sections reflects the number of samples that were included in the analysis, excluding removed observations and dropouts. Of the remaining 180 participants, 83 (46%) were given escitalopram monotherapy for the entire duration of the study, while 97 (54%) were augmented with aripiprazole in phase II. Further demographic and clinical information on the participants, including response rates, can be found in **Table 1**.”*

Additionally, the main manuscript now also includes a diagram of the clinical trial design and analytical workflow (**Figure 1**, previously in the supplementary material), which shows the number of participants in each arm of the clinical trial.

3. Some of the n's included in the results are inconsistent with the methods. Please clarify. For example:

- a. Page 10, line 198, the n for urinary outcomes, it lists the n as 314 instead of 323.
- b. Page 10, line 205, the n for depression outcomes is 172 instead of 211.

The *n* mentioned in the results reflects the number of samples that were included in the analysis. This is lower than what showed in the demographics table, due to samples that were excluded due to not satisfying quality control checks or due to dropouts before response data was collected at week 8/week 16. We have added a statement in the methods (**page 6, lines 95-99**) to clarify this point:

*“Plasma data from 12 participants and urine data from 9 participants were excluded as they did not pass quality control checks. In addition, 31 participants dropped out of the study before termination of the clinical trial. The *N* given in the following sections reflects the number of samples that were included in the analysis, excluding removed observations and dropouts.”*

4. In phase 2, presumably “baseline” refers to week 8 (the baseline for phase 2). If so, this might be clarified. Also, is “response” in phase 2 based on a 50% reduction on the MADRS from week 8?

In the clinical trial, the definition of “response” is a 50% or greater reduction in MADRS score from baseline to week 8 for escitalopram monotherapy and from baseline to week 16 for aripiprazole augmentation. In our analysis, we considered MADRS reduction occurring in phase I only (baseline-week 8), phase II only (week 9-week 16), as well as the overall reduction in both phases (baseline-week 16). This choice was taken to isolate the associations with the combination of escitalopram and adjunctive aripiprazole that were independent of the first 8 weeks of escitalopram monotherapy. We now state this clearly in the methods (**page 14, lines 306-311**):

“At the end of phase I (week 8), subjects were classified as ‘responders’ to escitalopram monotherapy if they achieved a 50% or greater reduction in MADRS score relative to baseline. In phase II (week 9-week 16), responders were maintained on the same escitalopram dose, while non-responders received escitalopram augmented with aripiprazole (2-10 mg). Response to the combination of escitalopram and adjunctive aripiprazole was defined as a 50% or higher reduction in MADRS score from baseline to week 16.”

And **page 17, lines 392-396**:

“Weighted correlation networks were constructed to identify clusters of highly connected metabolites explaining the variability in (i) MADRS score at baseline, (ii) the percentage MADRS reduction during phase I (between baseline and week 8), phase II (between week 8 and week 16) and during the entire trial (between baseline and week 16), as well as in the covariates of interest (i.e. age, BMI and CRP).”

5. Given that there were no metabolic predictors to escitalopram in the females, but there were metabolic predictors in phase 2, it would be interesting to know whether there were any significant changes in the metabolites from baseline to week 8, particularly among the responding males and non-responding females. In fact, a table with the means (SD) for baseline, week 8, and week 16 results for the depressed sample by gender and response status would be helpful.

Thank you for this suggestion. Generating such a large table is not feasible, due to the large number of features (112 lipoproteins), timepoints (baseline, week 8, week 16) and groups (male ESC responders, male ESC non-responders, female ESC responders, female ESC non-responders, male ESC+ARI responders, etc.). Given that we have already included fold-change information for the significant lipoproteins on both sexes (see **Supplementary Data 3 and 4**), we do not think this additional table would improve the quality of our manuscript. In addition, the data used for our analysis will be available through a public repository (Brain-CODE), allowing readers to have access to this information if interested in further analysis.

We thank you again for considering publication of our manuscript and we look forward to hearing back from you.

Kind regards,

Giorgia Caspani, Gustavo Turecki, Raymond W. Lam, Roumen V. Milev, Benicio N. Frey, Glenda M. MacQueen, Daniel J. Müller, Susan Rotzinger, Sidney H. Kennedy, Jane A. Foster, Jonathan Swann